# Bacterial contamination of surgical instruments in 25-gauge pars plana vitrectomy: Incidence and risk factors

Fumiya Miyako[1], Yosuke Harada[1*], Tomona Hiyama[1], Seiya Kashiyama[2], Saori Kawamoto[1], Chika Yokota[1], Akira Minamoto[1], Hiroki Ota[1], Ayako Sadahide[1], Kazuyuki Hirooka[1], Taiichiro Chikama[1], Hirokazu Sakaguchi[1]

**1** Department of Ophthalmology and Visual Science, Hiroshima University Graduate School of Biomedical Sciences, Hiroshima, Japan, **2** Department of Clinical Support, Clinical Laboratory Division, Hiroshima University, Hiroshima, Japan

* yharada@hiroshima-u.ac.jp

## Abstract

### Purpose

To evaluate the incidence of surgical instrument contamination associated with micro-incision vitrectomy surgery.

### Methods

Ninety-six eyes from 86 immunocompetent patients who underwent 25-gauge pars plana vitrectomy from December 2023 to February 2024 in our institution were retrospectively analyzed.

### Results

Surgical instrument contamination was detected in 7 cases (7.3%). Bacterial species were isolated from the trocar, endoilluminator, and beveled probe. Postoperative endophthalmitis did not occur during the 2-month follow-up period. Use of preoperative antibacterial eye drops was significantly associated with lower risk of bacterial contamination.

### Conclusion

Contamination may occur 25-gauge pars plana vitrectomy, even when preoperative antibacterial eye drops are used in conjunction with povidone-iodine irrigation and adequate vitreous dissection.

**Data availability statement:** The data underlying this study cannot be made publicly available due to ethical restrictions related to patient confidentiality. De-identified data are available from the Ethics Committee of Hiroshima University Hospital for researchers who meet the criteria for access to confidential data. Requests for data access should be directed to Dr. Narai at Hiroshima University Hospital (Department of Ophthalmology) (snarai@hiroshima-u.ac.jp). The data will be securely stored at Hiroshima University to ensure long-term availability.

**Funding:** The author(s) received no specific funding for this work.

**Competing interests:** The authors have declared that no competing interests exist.

## Background

Postoperative endophthalmitis is the most serious complication of vitrectomy and can result in loss of vision. Since Fuji et al. introduced the 25-gauge (G) transconjunctival sutureless vitrectomy system in 2002 [1], microincision vitrectomy surgery (MIVS) has become widely used. Although MIVS offers several advantages, it carries a theoretical risk of increased bacterial contamination owing to its transconjunctival approach. Even when meticulous preoperative precautions are taken, including administration of antimicrobial eye drops and disinfection with povidone-iodine, microorganisms can still be detected in the conjunctival sac both during and after the procedure [2,3]. Consequently, endophthalmitis after MIVS is thought to originate from the sclerotomy site and then extend into the vitreous space.

Although two previous studies reported that the incidence of postoperative endophthalmitis is higher following 25-gauge pars plana vitrectomy (PPV) compared to 20-gauge PPV [4,5], these investigations focused on clinically diagnosed infections and did not evaluate the presence of bacterial contamination on surgical instruments. While such contamination does not always lead to overt endophthalmitis, it may serve as an important source of postoperative infection. Therefore, understanding the frequency and nature of instrument contamination is critical for improving surgical safety and infection control, especially in the context of microincision vitrectomy. Strategies such as extensive vitrectomy with aggressive intraocular irrigation followed by complete sclerotomy sealing have been proposed to prevent postvitrectomy endophthalmitis [6,7]. Application of these strategies may have contributed to a decline in the incidence of MIVS-related endophthalmitis over the last decade [8–10]. However, postoperative endophthalmitis remains a clinically important complication.

Minimal vitrectomy and endoilluminator-assisted scleral buckling have gained widespread use [11–13] but also are associated with a risk of endophthalmitis. Therefore, assessment of bacterial contamination of the vitreous cavity is warranted. Although previous studies have used vitreous samples to detect bacterial contamination, contamination of surgical instruments and its role in MIVS-related endophthalmitis has not been extensively examined. This study aimed to investigate the incidence of surgical instrument contamination after 25-G PPV and determine factors associated with contamination.

## Methods

Patients who underwent 25-G PPV at Hiroshima University between December 2023 and February 2024 and were followed for at least 2 months after surgery were retrospectively reviewed. Medical records were retrospectively reviewed, and data were accessed for research purposes in December 2024. The authors had access to identifiable patient information during data collection, which was anonymized prior to analysis. This study was conducted in accordance with the tenets of the Declaration of Helsinki and was approved by the Institutional Review Board of Hiroshima University. Patients with ocular trauma, endophthalmitis, blepharitis, conjunctivitis, canaliculitis, lacrimal duct obstruction, or severe entropion were excluded. We also excluded

those on immunosuppressive agents and patients receiving chemotherapy. Bacterial cultures of surgical instruments (cannulas, beveled probes, endoilluminators) were obtained at the end of the operation.

PPVs were performed by vitreoretinal surgeons using the Constellation vitrectomy system (Alcon, Fort Worth, TX, USA). Local anesthesia was used for most cases; however, general anesthesia was used for patients who had difficulty remaining still during the operation. Preoperative antimicrobial eye drops, either levofloxacin (LVFX) or moxifloxacin (MFLX), were administered three times daily for 3 days, except in emergency situations. Simultaneous cataract surgery was performed for patients with existing cataracts or those over 50 years of age, as cataracts tend to progress after PPV. Before surgery, the skin around the eye was scrubbed with 10% povidone-iodine (Meiji Seika, Tokyo, Japan); the conjunctiva and cornea were disinfected with a 1:6 dilution of iodine: partially hydrolyzed polyvinyl alcohol (Rohto Nitten Co., Ltd., Nagoya, Japan). To prevent intraocular contamination, the ocular surface was irrigated with the same iodine/partially hydrolyzed polyvinyl alcohol solution just before creating the sclerotomy and again just before suturing the wound. For cases involving concomitant cataract surgery, the disinfection solution was also used just before intraocular lens implantation [14]. No additional repeated intraoperative iodine irrigation was performed.

All surgeries were performed by 5 vitreoretinal surgeons with varying levels of experience, ranging from less than one year of surgical experience to experienced vitreoretinal specialists with over 10 years of vitreoretinal surgical experience. For the analysis, surgeon experience was categorized into two groups (≥5 years and <5 years), and its association with contamination was assessed.

Cannulas were removed together with the endoilluminators or beveled probes by gently grasping them with toothed forceps and carefully withdrawing them through the scleral incision. This technique was part of our routine surgical practice for wound closure and was not specifically performed for the purpose of this study. Particular attention was paid to avoid contact with the conjunctiva or eyelid margin so as to minimize the risk of external contamination. Each cannula was then directly embedded into a blood agar plate. Although this method did not allow precise differentiation between contamination of the outer surface and the inner lumen, bacterial colony formation was observed in different patterns, including localization around the distal tip, the proximal hub, or both regions. To assess contamination of the endoilluminators and beveled probes, the intraocular portions of the instruments were pressed and inserted into the blood agar medium, thereby allowing potential microorganisms adherent to the surfaces to be cultured.

Bacterial cultures were performed using 5% sheep blood agar medium (Eiken Chemical Co., Ltd., Tokyo, Japan). The agar plates with the embedded surgical instruments were incubated aerobically at 37°C for 18–24 hours in a low-temperature incubator, and the presence of bacterial growth was assessed on the seventh postoperative day.

Antimicrobial susceptibility testing was performed using an automated susceptibility testing system (IA40MIC-i; Eiken Chemical Co., Ltd.), employing Dry Plate Eiken (broth microdilution method) corresponding to each identified bacterial species. The tests were conducted according to the manufacturer's instructions, and the minimum inhibitory concentrations (MICs) were determined. Antimicrobial susceptibility to 11 antibiotic agents (cefmenoxime, meropenem, gentamicin, erythromycin, azithromycin, chloramphenicol, vancomycin, ofloxacin, levofloxacin [LVFX], gatifloxacin, and moxifloxacin [MFLX]) was evaluated in accordance with the Clinical and Laboratory Standards Institute (CLSI) M100-S30 guidelines [15].

On the day after surgery, patients were instructed to administer antimicrobial eye drops (0.5% MFLX) and 0.1% betamethasone sodium phosphate (Shionogi Pharma Co., Ltd., Osaka, Japan) four times a day as well as bromfenac (Senju Pharmaceutical Co., Ltd., Osaka, Japan) two times a day and continue doing so for 3–4 weeks. In addition, tropicamide phenylephrine hydrochloride (Rohto Nitten Co., Ltd.) was administered twice daily for 1 week postoperatively to achieve mydriasis and reduce the risk of posterior synechiae formation.

Statistical analyses were conducted using JMP software version 18 (SAS Institute, Cary, NC, USA).

Because some patients contributed both eyes to the analysis, a generalized linear mixed model (GLMM) with patient ID as a random effect was used as the primary analytical approach to account for within-patient correlation. Given the limited

number of contamination events, we primarily performed univariable analyses. Odds ratios (ORs) with 95% confidence intervals (CIs) were calculated, and p values < 0.05 were considered statistically significant.

## Results

A total of 101 eyes underwent PPV during the study period. Five eyes were excluded from analysis owing to criteria (traumatic globe rupture, 3 eyes; bacterial blebitis, 2 eyes). Therefore, 96 eyes from 86 patients were analyzed, 10 patients underwent bilateral surgery with both eyes included.

Patient and clinical characteristics are shown in Table 1. Mean age was 64.7 ± 17.4 years. Thirty-three eyes were in men. Laterality was left in 44. The most common indications for PPV were rhegmatogenous retinal detachment (22 eyes), lens-related complications (19 eyes), proliferative diabetic retinopathy (16 eyes), and proliferative vitreoretinopathy (12 eyes). Thirty-three eyes underwent simultaneous phacoemulsification. The mean operation time was 69.4 ± 33.7 minutes. Nineteen eyes underwent standard three-port vitrectomy, whereas 77 eyes underwent four-port vitrectomy with chandelier illumination. Postoperative endophthalmitis did not occur within the 2-month follow-up period. Concomitant medical conditions included hypertension in 40 patients, diabetes mellitus in 29, and atopic dermatitis in 2; some patients had multiple comorbidities. Although 3 patients had a history of gastrointestinal cancer and 1 had a history of thyroid cancer, none were receiving chemotherapy. Eighty-five eyes (88.5%) received a prophylactic fluoroquinolone (LVFX in 61; MFLX in 24) before surgery. The remaining 11 eyes did not receive prophylaxis due to the emergency nature of the surgeries. Contamination was observed in 6 of 83 cases (7.2%) among surgeons with ≥5 years of experience, compared with 1 of 13 cases (7.7%) among those with <5 years of experience. (Table 1)

Bacteria were detected in the surgical instrument cultures of seven PPVs (7.3%). Fig 1 shows a representative blood agar medium plate on which bacteria were detected. The cannula was contaminated in 6 of the 7 contaminated cases. Among the contaminated cannulas, bacterial colonies were observed around the distal tip, the proximal hub, or both regions, without a clear predominant distribution pattern. The endoilluminator was contaminated in 1 and the vitrectomy probe was contaminated in 1. All bacteria isolated were gram-positive staphylococci. *Staphylococcus epidermidis* was isolated in 5 cases (71.4%); one of these isolates was methicillin resistant. *S. capitis* and *S. caprae* were isolated in the two remaining contaminated cases, respectively. None of the cultures grew multiple organisms. The isolated organisms and their antimicrobial susceptibility testing results are shown in Table 2. Three isolates (cases 4, 6, 7) demonstrated resistance to ofloxacin, LVFX, gatifloxacin, and MFLX.

The isolate of case 3 demonstrated resistance to ofloxacin, LVFX, and gatifloxacin but only intermediate resistance to MFLX. The four isolates exhibiting quinolone resistance included a methicillin-sensitive *S. epidermidis*, a methicillin-resistant *S. epidermidis*, a *S. capitis*, and a *S. caprae*.

In cases with contamination, intraoperative videos were reviewed to investigate potential causes. No instances of port dislodgement or reinsertion were observed in any of these cases. The characteristics of contaminated cases are summarized in Table 3. All procedures were performed by right-handed surgeons. Four cases were conducted as emergency surgeries, and preoperative antibacterial eye drops were not administered in these cases.

In a generalized linear mixed model, preoperative antibacterial eye drops and numbers of port were significantly associated with a lower risk of contamination Surgeon experience (≥5 vs < 5 years) was not significantly associated with contamination (Table 4)

## Discussion

Bacterial contamination of surgical instruments was identified in 7.3% of 25-G PPVs. Moreover, the use of preoperative antibacterial eye drops was significantly associated with lower risk of contamination. Contamination was predominantly found in cannulas; however, contamination of endoilluminators and vitrectomy probes also occurred.

**Table 1. Patient and clinical characteristics.**

|  | Overall characteristics (n=96) | Contaminated group (n=7) | Non-contaminated group (n=89) |
|---|---|---|---|
| Age (yrs), mean (SD) | 64,7 (17.4) | 64.29 (13.5) | 64.75 (17.8) |
| Gender (M/F) | 63/33 | 6/1 | 57/32 |
| Medical History |  |  |  |
| HT | 40 | 4 | 36 |
| DM | 29 | 3 | 26 |
| Atopic Dermatitis | 2 | 0 | 2 |
| Malignancies | 4 | 0 | 4 |
| Indication for PPV |  |  |  |
| RRD | 22 | 3 | 19 |
| PDR | 16 | 2 | 14 |
| PVR | 12 | 1 | 11 |
| Lens Dislocation | 5 | 1 | 4 |
| Aphakia | 7 | 0 | 7 |
| ERM | 7 | 0 | 7 |
| SRH | 6 | 0 | 6 |
| IOL Dislocation | 5 | 0 | 5 |
| Macular Hole | 5 | 0 | 5 |
| Vitreous Hemorrhage | 3 | 0 | 3 |
| VMTS | 2 | 0 | 2 |
| Nucleus drop | 2 | 0 | 2 |
| CCD | 1 | 0 | 1 |
| CH | 1 | 0 | 1 |
| Lamellar MH | 1 | 0 | 1 |
| Rubeotic Glaucoma | 1 | 0 | 1 |
| Ophthalmic Surgery Histories |  |  |  |
| Any type |  | 4 | 59 |
| PPV |  | 1 | 33 |
| Other than PPV |  | 26 | 3 |
| Surgical duration(minutes). Mean(SD) |  | 80.57(39.64) | 68.57(32.86) |
| Pre operative eye drop |  |  |  |
| LVFX | 61 | 3 | 58 |
| MFLX | 24 | 1 | 23 |
| None (emergency surgery) | 11 | 3 | 8 |
| Numbers or Port |  |  |  |
| Three-port | 19 | 4 | 15 |
| Four-port | 77 | 3 | 74 |
| surgeon experience (≧5/<5 years) | 83/13 | 6/1 | 77/12 |

HT, hypertension; DM, diabetes mellitus; RRD, rhegmatogenous retinal detachment; PVR, proliferative vitreoretinopathy; ERM, epiretinal membrane; SRH, subretinal hemorrhage; VMTS, vitreomacular tractional syndrome; CCD, ciliochoroidal detachment; CH, choroidal hemorrhage; PPV, pars plana vitrectomy.

Although bacterial contamination of surgical instruments was detected in 7.3% of cases, no postoperative endophthalmitis occurred in this study. This finding suggests that the presence of bacterial contamination does not necessarily lead to clinically apparent infection. A previous study has demonstrated that a minimum infectious inoculum is required for

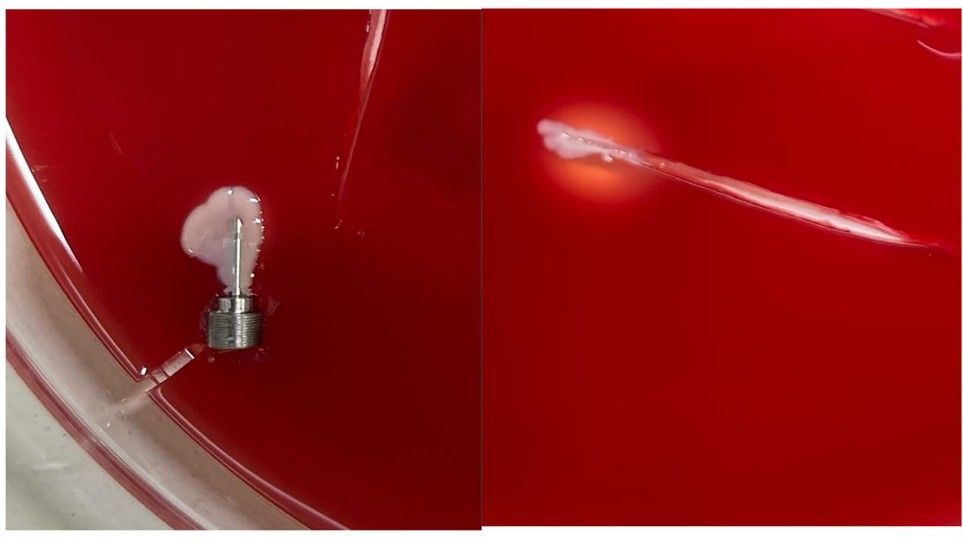

Fig1a          Fig1b

**Fig 1. Bacteria growing on blood agar medium surrounding a cannula and an endoilluminator. a)** Bacterial colonies were obsereved around a cannula. **b)** Bacterial colonies were observed growing around the area where the endoilluminator was inserted.

**Table 2. Bacteria isolated and antimicrobial sensitivity testing.**

| Case | Bacterial species | preoperative eye drop | anti bacterial drug and MIC | | | | | | | | | | | bacterial detected instruments |
|------|-------------------|-----------------------|-----|------|----|----|-----|----|-----|------|------|------|------|--------------------------------|
|      |                   |                       | CMX | MEPM | GM | EM | AZM | CP | VCM | OFLX | LVFX | GFLX | MFLX |                                |
| 1 | S.epidermidis | MFLX | S | S | S | S | S | S | S | S | S | S | S | 1 trocar/cannula |
| 2 | S.epidermidis | MFLX | S | S | S | S | S | S | S | S | S | S | S | endoilluminator |
| 3 | S.epidermidis | n | S | S | I | S | S | S | S | R | R | R | I | 1 trocar/cannula |
| 4 | S.epidermidis | LVFX | S | R | S | R | R | S | S | R | R | R | R | 2 trocar/cannulas |
| 5 | S.epidermidis | n | S | S | R | S | S | S | S | S | S | S | S | 1 trocar/cannula |
| 6 | S.capitis (MRS) | LVFX | S | S | S | S | S | S | S | R | R | R | R | 1 trocar/cannula beveled prove |
| 7 | S.caprae | n | S | S | R | S | S | S | S | R | R | R | R | 1 trocar/cannula |

CMX, cefmenoxime; MEPM, meropenem; GM, gentamicin; EM, erythromycin; AZM, azithromycin; CP, chloramphenicol; VCM, vancomycin, OFLX, oflox-acin; LVFX, levofloxacin; GFLX, gatifloxacin; MFLX, moxifloxacin.

the development of endophthalmitis, and small amounts of bacteria may be eliminated by host defense mechanisms or intraoperative irrigation [16]. Therefore, instrument contamination may represent a risk factor for postoperative infection. These results indicate that while contamination can occur during surgery, its clinical impact may depend on factors such as bacterial load, virulence, and intraoperative conditions. This may explain why endophthalmitis remains a rare com-plication despite the relatively frequent detection of bacterial contamination. In the present study, bacterial load was not quantified, and therefore, the relationship between detected contamination and the actual risk of postoperative infection

**Table 3. Characteristics of contaminated cases.**

| Case | Indication for PPV | Emergency surgery | Bacterial detected instruments | Numbers pf Port | Surgical duration(minutes) | Surgeon experience (years) | Ophthalmic Surgical History |
|------|-------------------|-------------------|-------------------------------|-----------------|---------------------------|---------------------------|----------------------------|
| 1 | RRD | Yes | 1 trocar/cannula | 3 | 63 | 10 | Cataract surgery |
| 2 | PVR | No | endoilluminator | 3 | 27 | 10 | PPV |
| 3 | Lens dislocation | No | 1 trocar/cannula | 3 | 162 | <1 | none |
| 4 | PDR | No | 2 trocar/cannulas | 3 | 55 | 10 | PPV |
| 5 | RRD | Yes | 1 trocar/cannula | 4 | 83 | 10 | none |
| 6 | PDR | No | 1 trocar/cannula beveled prove | 4 | 49 | 10 | Trabeculectomy |
| 7 | RRD | Yes | 1 trocar/cannula | 4 | 92 | 10 | Cataract surgery |

PPV, pars plana vitrectomy; RRD, rhegmatogenous retinal detachment; PVR, proliferative vitreoretinopathy; PDR, proliferative diabetic retinopathy.

**Table 4. Univariable generalized linear mixed model analysis of factors associated with contamination.**

| Variable | OR | 95% CI | P value |
|----------|-----|--------|---------|
| Age | 0.99 | 0.95-1.02 | 0.45 |
| Gender | 1.53 | 0.42-5.62 | 0.52 |
| Medical History | 1.21 | 0.31-4.68 | 0.78 |
| Indication for PPV | – | – | 0.84 |
| Ophthalmic Surgery Histories | | | |
| Any type | 0.78 | 0.21-2.89 | 0.71 |
| PPV | 0.34 | 0.07-1.69 | 0.19 |
| Other than PPV | 1.28 | 0.30-5.39 | 0.74 |
| Surgical duration | 1.02 | 0.99-1.03 | 0.32 |
| Pre operative eye drop | 0.13 | 0.025–0.695 | 0.017 |
| Surgeon experience (≥5 vs<5 years) | 0.94 | 0.10–8.43 | 0.95 |
| Numbers of Port | 6.58 | 1.38-31.31 | 0.018 |

OR Odds ratio; CI, Confidence Interval; PPV, Pars plana Vitrectomy.

remains unclear. Quantitative assessment of bacterial burden, such as colony-forming unit counts, would be valuable in future studies to better understand the clinical significance of contamination.

MIVS may pose a risk of transconjunctival bacterial contamination from ocular surface flora, unlike 20-G PPV [6]. In a study of irrigation methods in 25-G vitrectomy, coagulase-negative *staphylococci* were present in 2% of patients who received infusion fluid irrigation and in 0.6% of patients who received povidone iodine irrigation at the end of vitrectomy [14]. These findings suggest that the risk of bacterial contamination in the vitreous cavity might be reduced by performing extensive vitrectomy or using povidone iodine irrigation.

Data regarding contamination of surgical instruments that penetrate the vitreous cavity remains limited. In the present study, a contamination rate of 7% was observed when culturing instruments at the end of surgery, suggesting that contamination may occur during intraoperative instrument handling. The observed distribution of bacterial growth on cannulas varied, with colonies detected at the distal tip, the proximal hub, or both regions. This finding suggests that contamination may occur through multiple routes, including contact with the ocular surface or surgical field as well as potential entry

through the cannula lumen during intraoperative manipulation. Although no port dislodgement or reinsertion was observed in the contaminated cases, contamination may still occur through other mechanisms. The precise mechanisms of contamination could not be determined in this study. However, several possible pathways may be considered. For example, surgical instruments such as light probes or vitrectomy probes may come into contact with pooled irrigation fluid around the operative field or on the surgical drape and subsequently be reinserted into the eye. In addition, repeated insertion and removal of instruments through the cannula may increase the likelihood of contamination. These potential mechanisms warrant further investigation, ideally with detailed intraoperative video analysis. The number of instrument insertions through the port may also influence the risk of contamination, as repeated insertion and removal of instruments could increase the opportunity for bacterial entry. However, this variable was not recorded in the present study, and further investigations are needed to clarify its impact. Interestingly, contamination was more frequently observed in three-port procedures than in four-port procedures. (Tables 1, 4) This finding may be explained by differences in surgical technique. In four-port surgery, the use of chandelier illumination allows for more stable visualization and reduces the need for repeated instrument insertion and removal, which may decrease the risk of contamination. In contrast, three-port procedures often require more frequent instrument exchanges, potentially increasing the likelihood of contact with the ocular surface, surgical drape, or surrounding fluid. These findings suggest that the frequency of instrument manipulation, rather than surgical duration itself, may play a more important role in contamination. However, this association should be interpreted with caution, as it may be influenced by confounding factors such as differences in case selection and surgical complexity.

Considering that newer surgical techniques such as minimal posterior pole vitrectomy [11] and endoilluminator-assisted scleral buckling [12,13] leave the entire vitreous or at least a significant portion remaining, ophthalmologists must be wary of the risk of postoperative infection when performing them.

Achieving complete elimination of bacteria from the surgical field remains a challenge in MIVS. Preoperative LVFX eye drops combined with conjunctival irrigation of the fornixes using 5% povidone-iodine is effective in reducing conjunctival bacterial flora. [2,3] However, even with repeated flushing of the operative field with povidone-iodine, these measures cannot completely prevent bacterial invasion [6,14,17]. Our study demonstrated that use of preoperative LVFX or MFLX eye drops significantly lowered the incidence of surgical instrument contamination; however, contamination was not entirely eliminated. Moreover, despite flushing the operative field with povidone-iodine, some vitrectomy probes and endoilluminators were contaminated. We suspect that these instruments became contaminated upon direct contact with the conjunctiva.

Several studies have reported a global increase in fluoroquinolone-resistant bacteria [18,19]. In particular, Kim et al. [18] reported that repetitive use of fluoroquinolone eye drops in patients undergoing intraocular injection led to an increase in isolation of fluoroquinolone-resistant bacteria [19]. We exclusively used fluoroquinolone antibacterial eyedrops in our study, which may explain why 4 of the 7 bacteria isolates detected were fluoroquinolone-resistant. In our study, 4 of 7 isolates demonstrated fluoroquinolone resistance. However, given the small number of contamination events, these findings should be interpreted with caution. While this observation may suggest a potential concern regarding fluoroquinolone resistance, further studies with larger sample sizes are required before drawing conclusions regarding changes in preoperative antibiotic strategies.

Vitreous sampling is considered the gold standard for pathogen identification in cases of endophthalmitis. However, the diagnostic yield of conventional cultures is only approximately 50%, which increases to up to 70% when hemoculture bottles are employed [20–24]. In our study, bacteria were successfully detected on surgical instruments and antimicrobial susceptibility was assessed. By culturing the surgical instruments themselves, the potential pathogens responsible for postoperative endophthalmitis can be identified more readily. This approach offers an important advantage: if postoperative endophthalmitis develops, the likely causative organism can be predicted immediately, enabling prompt and targeted treatment.

Moreover, in cases where bacterial contamination is detected on surgical instruments, postoperative management strategies can be adapted. For instance, clinicians may choose to extend hospitalization or increase the frequency of

 

outpatient visits, thereby facilitating earlier recognition of clinical signs and ensuring preparedness for immediate intervention should endophthalmitis occur.

This strategy is particularly valuable in the context of low-virulence organisms such as *Cutibacterium Acnes* [ 25], which may cause endophthalmitis with subtle symptoms, delayed onset, and a slowly progressive course. When such organisms are identified on surgical instruments, clinicians can adjust postoperative follow-up accordingly. For example, scheduling more frequent and prolonged outpatient visits. Thereby improving the chances of timely detection and effective management of infection. However, considering the low incidence of postoperative endophthalmitis and the associated costs, routine culturing of surgical instruments in all cases may not be practical. Rather, a targeted approach focusing on higher-risk situations—such as cases without preoperative antibiotic eye drops—may be more appropriate. Although routine culturing remains debatable, these findings suggest that culturing surgical instruments may be a valuable adjunct for selecting antimicrobial agents, especially when vitreous samples yield negative results.

In this study, patients receiving immunosuppressive agents or chemotherapy were excluded. However, such patient-related factors may influence the risk of bacterial contamination during surgery. Future studies including these populations would be valuable to further clarify their impact.

## Limitation

This study has several limitations. 1; microbiological cultures of surgical instruments were not performed at the initiation of the procedure. However, since the surgeons followed appropriate aseptic preparations, the risk of contamination before the start of surgery is considered minimal. 2; conjunctival sac cultures were not obtained, which limits our ability to directly compare the bacterial flora of the ocular surface with the organisms detected on surgical instruments. 3; this was a single-center study conducted under specific surgical techniques and perioperative protocols, and that the generalizability of the findings may be limited. Further large-scale studies are needed to confirm our findings. 4; although the cannulas, endoilluminators, and beveled probes were handed as cleanly as possible immediately after surgery, it is possible that the detected bacteria were contaminants from outside the operative field. 5; we did not conduct polymerase chain reaction or culture testing of ocular surface fluid and vitreous samples, nor were surgical instruments cultured at the beginning of surgery. 6; The number of contamination events in this study was relatively small (n = 7), which may limit the stability and reliability of the statistical analysis. In particular, the low number of events per variable raises the possibility of model overfitting and model instability, which may have contributed to the wide confidence intervals observed. Therefore, the results should be interpreted with caution. 7; It should also be noted that cannulas were removed together with the endoilluminators or beveled probes as part of the standard surgical technique. During this process, there is a potential risk of cross-contamination, as bacteria attached to one instrument may come into contact with another. Therefore, the possibility that such interactions may have influenced the culture results cannot be completely excluded. 8; The positional information of each instrument or cannula was not specifically recorded in this study; therefore, we were unable to evaluate the association between surgical quadrant and the risk of contamination.

## Conclusion

Bacteria were detected from cannulas, endoilluminators, and beveled probes cultured at the end of surgery in 7.3% of 25-G PPVs. The use of preoperative antibacterial eye drops was significantly associated with a lower risk of contamination; however, bacterial contamination of surgical instruments may still occur despite standard preventive measures, including preoperative antibacterial eye drops, povidone-iodine irrigation, and adequate vitreous dissection. The presence of fluoroquinolone-resistant bacterial strains observed in this study may warrant further investigation; however, conclusions regarding changes in preoperative antibiotic strategies should be interpreted with caution due to the limited sample size and the absence of postoperative endophthalmitis in this study.

## Author contributions

**Conceptualization:** Yosuke Harada.

**Data curation:** Fumiya Miyako, Tomona Hiyama, Seiya Kashiyama.

**Formal analysis:** Fumiya Miyako.

**Investigation:** Fumiya Miyako.

**Methodology:** Yosuke Harada.

**Resources:** Fumiya Miyako.

**Software:** Fumiya Miyako.

**Supervision:** Yosuke Harada, Tomona Hiyama, Hirokazu Sakaguchi.

**Validation:** Fumiya Miyako.

**Visualization:** Fumiya Miyako.

**Writing – original draft:** Fumiya Miyako.

**Writing – review & editing:** Yosuke Harada, Tomona Hiyama, Saori Kawamoto, Chika Yokota, Akira Minamoto, Hiroki Ota, Ayako Sadahide, Kazuyuki Hirooka, Taiichiro Chikama, Hirokazu Sakaguchi.

## Acknowledgments

We thank Edanz (https://jp.edanz.com/ac) for editing a draft of this manuscript.

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
