## [Decision Letter · Decision Letter 0]

16 Mar 2026

PONE-D-25-68319Bacterial Contamination of Surgical Instruments in 25-Gauge Pars Plana Vitrectomy: Incidence and Risk FactorsPLOS One

Dear Dr. MIyako

Thank you for submitting your manuscript to PLOS ONE. After careful consideration, we feel that it has merit but does not fully meet PLOS ONE’s publication criteria as it currently stands. Therefore, we invite you to submit a revised version of the manuscript that addresses the points raised during the review process. Please submit your revised manuscript by Apr 30 2026 11:59PM. If you will need more time than this to complete your revisions, please reply to this message or contact the journal office at plosone@plos.org. Please include the following items when submitting your revised manuscript:

We look forward to receiving your revised manuscript.

Kind regards,

Jiro Kogo

Academic Editor

PLOS One

Journal Requirements:

3. In the online submission form you indicate that your data is not available for proprietary reasons and have provided a contact point for accessing this data. Please note that your current contact point is a co-author on this manuscript. According to our Data Policy, the contact point must not be an author on the manuscript and must be an institutional contact, ideally not an individual. Please revise your data statement to a non-author institutional point of contact, such as a data access or ethics committee, and send this to us via return email. Please also include contact information for the third party organization, and please include the full citation of where the data can be found.

Reviewers' comments:

Reviewer's Responses to Questions

**Comments to the Author**

1. Is the manuscript technically sound, and do the data support the conclusions?

Reviewer #1: Yes

Reviewer #2: Partly

Reviewer #3: Yes

2. Has the statistical analysis been performed appropriately and rigorously? 

Reviewer #1: Yes

Reviewer #2: No

Reviewer #3: Yes

3. Have the authors made all data underlying the findings in their manuscript fully available?

Reviewer #1: Yes

Reviewer #2: No

Reviewer #3: Yes

4. Is the manuscript presented in an intelligible fashion and written in standard English?

Reviewer #1: Yes

Reviewer #2: Yes

Reviewer #3: Yes

5. Review Comments to the Author

Reviewer #1: This manuscript evaluates bacterial contamination of surgical instruments following 25-gauge pars plana vitrectomy and identifies preoperative antibacterial eye drops as a factor associated with reduced contamination. The topic is clinically relevant, and the direct culturing of instruments represents a practical and interesting approach. The manuscript is generally well organized and clearly written.

I have a few comments that may strengthen the paper:

1. Although a 7.3% contamination rate was observed, no cases of postoperative endophthalmitis occurred. A brief discussion clarifying the clinical implications of instrument contamination in the absence of infection would be helpful.

2. The number of contamination events is small (n = 7). Please comment on the stability of the multivariable analysis and consider acknowledging this limitation more explicitly.

3. Four of seven isolates demonstrated fluoroquinolone resistance. While important, the sample size is limited; therefore, conclusions regarding changes in preoperative antibiotic strategy should be presented cautiously.

Overall, this is a meaningful study addressing an underreported aspect of microincision vitrectomy. With minor clarifications and a slightly expanded discussion, the manuscript would be strengthened.

Reviewer #2: I appreciate the opportunity to review this manuscript investigating bacterial contamination of surgical instruments during 25-gauge pars plana vitrectomy. The topic is clinically relevant, as postoperative endophthalmitis remains a serious complication of vitreoretinal surgery. Evaluating potential contamination routes during surgery is therefore important. The concept is interesting; however, several methodological and statistical issues should be addressed before further consideration. Overall, the conclusions and discussion, particularly regarding the clinical implications, appear somewhat overstated and the tone should be moderated.

・Because some patients contributed both eyes to the analysis, the observations are not statistically independent. Although the authors applied a generalized linear mixed model (GLMM) with patient ID as a random effect, the initial variable screening was performed using standard multivariable logistic regression at the eye level. It would be more appropriate to use the GLMM as the primary analysis from the outset.

・In addition, because the number of contamination events is very small in this study, multivariable logistic regression may be prone to overfitting and may not be statistically appropriate. The authors may consider limiting the analysis to univariable logistic regression and acknowledging in the limitations that multivariable analysis could not be reliably performed due to the limited number of events. The manuscript would also benefit from clearer reporting of how many patients underwent bilateral surgery and whether both eyes were included in the analysis.

・The conclusion that contamination is “not completely avoidable” during 25-gauge pars plana vitrectomy may be somewhat overstated. Because this study was conducted at a single institution with specific surgical techniques and perioperative protocols, the findings may not necessarily be generalizable to all surgical settings.

Differences in surgical technique, operating room environment, infection control protocols, and surgeon experience may influence contamination rates. Therefore, the authors may wish to moderate the tone of the conclusion and state that contamination may occur during 25-gauge PPV rather than suggesting that it is inherently unavoidable.

・The manuscript mentions the use of Phisoderm solution in patients with iodine allergy. Please provide a more detailed description of this solution (e.g., its active components). It would also be helpful to clarify whether this agent contains chlorhexidine or another antiseptic. In addition, please report the number or proportion of patients in whom this solution was used.

・The authors state that tropicamide phenylephrine hydrochloride was administered twice daily for 3–4 weeks after surgery. Please clarify the rationale for this postoperative treatment. (L162-163)

・The Methods section should include information regarding the surgeons who performed the procedures and their level of surgical experience. Surgeon experience may influence the risk of intraoperative contamination. In complex cases such as PVR, PDR, and RRD, which actually required longer operative times and advanced surgical skills, the risk of contamination during surgery may be higher.

・It would be important to clarify how many ports were used during the vitrectomy procedures. Specifically, please indicate whether the surgeries were performed using a standard three-port system or whether a four-port system with chandelier illumination was used in some cases.

・The discussion of COVID-19–related changes in endophthalmitis incidence in the Introduction may not be directly relevant to the main focus of this study. The authors may wish to reconsider whether this information is necessary in the Introduction.

・Based on the Methods section, povidone-iodine irrigation appears to have been applied before trocar insertion; therefore, bacterial entry into the eye at the time of port insertion seems relatively unlikely. It would therefore be important to clarify where bacterial growth was most frequently detected on the cannulas. For example, was bacterial growth mainly detected on the outer surface of the cannula extending toward the tip, or within the inner lumen of the cannula? If possible, the distribution of bacterial growth along the cannula should be described.

・On the other hand, in complex cases such as PVR, longer surgical duration and repeated intraoperative manipulations may increase the possibility of contamination. In the present study, contamination occurred in cases with RRD, PDR, and PVR, in which the surgical duration also appears to have been relatively longer. For example, during surgery the port may become partially dislodged and require reinsertion, or the surgeon may need to push the cannula back into position. Such additional manipulations could potentially introduce contamination. Therefore, it would be helpful if the authors could clarify whether any ports became dislodged and required reinsertion during surgery, and report the number of such events if they occurred. This information would help readers better interpret the potential mechanisms of contamination observed in this study.

・In particular, for the cases in which contamination was detected on surgical instruments such as the vitrectomy probe or cannula, it would be valuable to review the intraoperative videos, if available, to investigate the potential cause of contamination in greater detail. If the cause cannot be clearly identified even after such review, the authors should consider discussing other possible mechanisms in the Discussion section.

For example, contamination might occur if an instrument such as the light probe comes into contact with pooled irrigation fluid around the operative field or on the surgical drape and is then reinserted into the eye. Exploring such potential mechanisms would help readers better understand how contamination may occur during surgery.

・It would also be clinically useful to evaluate whether contamination was associated with the number of instrument insertions through the port during surgery.

・Please clarify the rationale for the procedure described in lines 138–141. If a light guide or similar device is inserted into the cannula, there is a possibility that bacteria could adhere to it. Even at this stage, bacteria attached to the endoilluminators, beveled probes, or trocars could potentially come into contact with each other, leading to cross-contamination. The authors should discuss whether this possibility may have affected the culture results.

・Previous studies by Shimada H (The author of Reference 7). and colleagues have many times reported that repeated povidone-iodine irrigation can reduce the risk of bacterial entry into the vitreous cavity. Please clarify whether such repeated iodine irrigation was performed in the present study.

・More detailed information about the contaminated cases would be helpful. For example, among the cases of RRD, PDR, PVR, and lens dislocation, please specify which surgical instruments were contaminated. Presenting these data in a table would improve clarity. In addition, it would be useful to evaluate whether contamination was associated with factors such as laterality, surgeon handedness, or surgical quadrant. The manuscript also does not specify how many of these cases were emergency surgeries, which should be clarified.

・The discussion in lines 274–291 suggests that routine culturing of surgical instruments may be beneficial. However, considering the very low incidence of postoperative endophthalmitis and the associated costs, routine culturing for all cases may not be practical in clinical settings. Based on the findings of this study, it may be more appropriate to focus the discussion on identifying higher-risk situations, such as cases in which preoperative antibiotic eye drops were not administered, rather than emphasizing the general usefulness of routine instrument cultures.

・Table 1 appears to contain an error in the age values of the non-contaminated group. The reported mean age and standard deviation (“19.43 (3.57)”) seem inconsistent with the overall cohort and likely represent a typographical error.

・Minor language and typographical issues should be corrected. For example, “obsereved” in the figure legend should be corrected to “observed”.

・It may also be informative to include patients receiving immunosuppressive agents or chemotherapy in future studies to evaluate whether these conditions influence the risk of surgical instrument contamination. (L109-110)

Reviewer #3: The authors investigated bacterial contamination associated with cannulas during PPV by performing bacterial cultures of surgical instruments. They detected bacterial contamination in 7.3% of cases (7/96 eyes) and identified the use of preoperative antibacterial eye drops as a factor associated with contamination in the multivariable logistic regression analysis.

This study provides valuable data regarding the incidence of bacterial contamination during MIVS. However, several critical concerns remain.

Issue 1. Interpretation of contamination versus clinical infection

A. Clinical relevance

Although bacterial contamination was detected in 7.3% of cases, no postoperative endophthalmitis occurred during the follow-up period. Therefore, the clinical significance of instrument contamination remains unclear.

The manuscript introduces postoperative endophthalmitis as a major concern in the Background section; however, the relationship between the detected contamination and the risk of endophthalmitis is insufficiently discussed.

In particular, the statement in the Conclusion that“The considerable rise in fluoroquinolone-resistant bacterial strains suggests that ophthalmologists should reassess their selection of preoperative antibacterial agents” appears to go well beyond what can be supported by the data presented in this study.

B. Relationship with bacterial load

Experimental studies using rabbit endophthalmitis models have suggested that intraocular infection typically requires a minimum infectious inoculum, often approximately 10²–10³ CFU depending on the bacterial species.

The mere detection of bacteria does not necessarily indicate a clinically meaningful infection risk. It would be informative to evaluate the bacterial load associated with the detected contamination, as well as its relationship with preoperative or intraoperative factors. Thus, quantification of bacterial burden (e.g., CFU counts) would help clarify whether the detected contamination could plausibly lead to postoperative endophthalmitis.

Issue 2. Sample size and statistical power

Although the sample size of 96 eyes is adequate for estimating the contamination rate, the number of contamination events (7 cases) is very small for robust multivariable analysis. With such a limited number of events, the multivariable logistic regression and GLMM models may be unstable. The events-per-variable assumption (EPV < 10) is likely not satisfied, raising concerns about model overfitting. This concern is reflected in the wide confidence interval reported for the association between preoperative antibacterial eye drops and contamination (OR 0.13, 95% CI: 0.025–0.695).

Given the small number of contamination events, the robustness of the statistical analysis may be limited. The authors should explicitly acknowledge this limitation in the manuscript and interpret the results with caution. Alternatively, increasing the sample size or number of events in future studies would help strengthen the reliability of the findings.

6. PLOS authors have the option to publish the peer review history of their article (what does this mean?). If published, this will include your full peer review and any attached files.

Reviewer #1: **Yes:** Yoshihito Sakanishi

Reviewer #2: No

Reviewer #3: No

---

## [Author Response · Author response to Decision Letter 1]

16 Apr 2026

Dear Editor and Reviewers,

We sincerely thank the Academic Editor and the reviewers for their thorough evaluation of our manuscript and for their valuable and constructive comments. We greatly appreciate the time and effort devoted to reviewing our work.

---

## [Decision Letter · Decision Letter 1]

27 Apr 2026

PONE-D-25-68319R1Bacterial Contamination of Surgical Instruments in 25-Gauge Pars Plana Vitrectomy: Incidence and Risk FactorsPLOS One

Dear Dr. Miyako,

Thank you for submitting your manuscript to PLOS ONE. After careful consideration, we feel that it has merit but does not fully meet PLOS ONE’s publication criteria as it currently stands. Therefore, we invite you to submit a revised version of the manuscript that addresses the points raised during the review process.

We look forward to receiving your revised manuscript.

Kind regards,

Jiro Kogo

Academic Editor

PLOS One

Journal Requirements:

Reviewers' comments:

Reviewer's Responses to Questions

**Comments to the Author**

1. If the authors have adequately addressed your comments raised in a previous round of review and you feel that this manuscript is now acceptable for publication, you may indicate that here to bypass the “Comments to the Author” section, enter your conflict of interest statement in the “Confidential to Editor” section, and submit your "Accept" recommendation.

Reviewer #1: All comments have been addressed

Reviewer #2: (No Response)

Reviewer #3: All comments have been addressed

2. Is the manuscript technically sound, and do the data support the conclusions?

Reviewer #1: Yes

Reviewer #2: Partly

Reviewer #3: Yes

3. Has the statistical analysis been performed appropriately and rigorously? 

Reviewer #1: Yes

Reviewer #2: Yes

Reviewer #3: Yes

4. Have the authors made all data underlying the findings in their manuscript fully available?

Reviewer #1: Yes

Reviewer #2: No

Reviewer #3: Yes

5. Is the manuscript presented in an intelligible fashion and written in standard English?

Reviewer #1: Yes

Reviewer #2: No

Reviewer #3: Yes

6. Review Comments to the Author

Reviewer #1: The authors have responded appropriately and thoroughly to the reviewer’s comments, and the manuscript is very well written overall. Each point raised has been carefully addressed with clear and sufficient explanations, demonstrating the authors’ strong understanding of the subject and their commitment to improving the quality of the work. The revisions have significantly enhanced the clarity, consistency, and scientific rigor of the manuscript. In particular, the additional details and clarifications provided in response to the comments have improved the transparency and reproducibility of the study. The structure is logical, and the arguments are presented in a coherent and persuasive manner. Overall, the manuscript is now in excellent condition and suitable for publication.

Reviewer #2: I appreciate the opportunity to review this revised manuscript. The authors have addressed many of the previous comments, particularly regarding the statistical analysis and clarification of surgical procedures, which has improved the manuscript.

However, some issues remain insufficiently addressed, including surgeon-related factors and the interpretation of results, which appears somewhat overstated, particularly in the Abstract. Addressing these points would be needed.

・The role of surgeon-related factors remains insufficiently addressed. Please include the surgeons’ years of experience in Table 4.

・In addition, the manuscript does not clearly specify how many surgeons were involved in this study. This information should be explicitly reported.

・It would also be valuable to evaluate the association between surgeon experience and contamination in Tables 1 and 3. If analysis as a continuous variable is difficult, categorizing surgeons into groups (e.g., experienced vs. less experienced, such as ≥4–5 years) may be more appropriate. Furthermore, the variable “Surgeons” in Table 3 is unclear and should be clarified.

・Throughout the manuscript, there are too many short paragraphs. Consolidating them into more meaningful and structured paragraphs would improve readability, particularly in the Discussion section.

・The conclusion that contamination is “not completely avoidable” appears not to have been revised in the Abstract. This statement should be moderated to better reflect the limitations of the study.

Reviewer #3: Manuscript has been improved with the additional remarks on strengths and limitations of this study. I have no further comment.

7. PLOS authors have the option to publish the peer review history of their article (what does this mean?). If published, this will include your full peer review and any attached files.

Reviewer #1: No

Reviewer #2: No

Reviewer #3: No

---

## [Author Response · Author response to Decision Letter 2]

1 May 2026

Dear Editor and Reviewers,

We sincerely thank the Academic Editor and the reviewers for their valuable comments and suggestions. We have carefully revised the manuscript in accordance with all comments. Detailed, point-by-point responses are provided in the attached document entitled “Response to Reviewers.”

We respectfully submit the revised manuscript for your consideration.

---

## [Editor Report · Decision Letter 2]

4 May 2026

Bacterial Contamination of Surgical Instruments in 25-Gauge Pars Plana Vitrectomy: Incidence and Risk Factors

PONE-D-25-68319R2

Dear Dr. Harada

We’re pleased to inform you that your manuscript has been judged scientifically suitable for publication and will be formally accepted for publication once it meets all outstanding technical requirements.

Kind regards,

Jiro Kogo

Academic Editor

PLOS One
---

## [Editor Report · Acceptance letter]

PONE-D-25-68319R2

PLOS One

Dear Dr. Harada,

I'm pleased to inform you that your manuscript has been deemed suitable for publication in PLOS One. Congratulations! Your manuscript is now being handed over to our production team.

Kind regards,

on behalf of

Prof. Jiro Kogo

Academic Editor

PLOS One